# Score-based Lyapunov Stable Neural ODE for Robust Classification

## Abstract

Adversarial attacks pose a significant obstacle to the widespread deployment of modern AI systems. These attacks, often implemented as imperceptible perturbations to the input, typically an image, can deliberately mislead neural networks into making incorrect predictions. Over the past decade, numerous studies have sought to understand and mitigate this vulnerability. Among them, a promising line of research interprets neural networks as dynamical systems and leverages Lyapunov theory to enhance robustness against adversarial perturbations. However, the original intent of Lyapunov theory differs from that of building accurate and robust neural networks, leading to conceptual and practical challenges. Existing approaches typically incorporate Lyapunov constraints through penalization, but such formulations only ensure local stability around input data points and do not guarantee broader regions of convergence. In this work, we propose a framework based on vector fields that explicitly admit asymptotically stable equilibrium points, thereby strengthening the Lyapunov-based foundation of the model. This enhanced theoretical grounding enables us to prove that every point within the support of the input data distribution converges to a stable equilibrium point, and enables us to draw a natural connection to the concept of score estimation. Experimentally, we demonstrate that our model improves adversarial robustness over prior Lyapunov-regularized approaches across standard image classification benchmarks. Qualitatively, the induced dynamics exhibit a denoising effect against adversarial perturbations, driving inputs toward stable modes of the data distribution.

## 1  Introduction

Adversarial attacks remain a fundamental challenge to the reliable deployment of modern AI systems (Apruzzese et al., 2023). These attacks, consisting of imperceptible perturbations crafted to induce misclassification, expose the fragility of deep networks and raise concerns about their use in safety-critical and ethically sensitive domains such as autonomous driving, biometric authentication, and medical diagnosis. Conventional defense strategies, notably adversarial training (Madry et al., 2018; Goodfellow et al., 2015), augment datasets with adversarial examples to enhance robustness. However, this approach is highly dependent on the specific attack used during training, often leads to poor generalization to unseen attacks, and typically degrades clean accuracy, a phenomenon known as the robustness-accuracy tradeoff (Tsipras et al., 2019; Zhang et al., 2019; Raghunathan et al., 2020). This has motivated a complementary line of research that seeks *structural robustness*: designing neural architectures whose dynamics are inherently stable to small perturbations, even without exposure to adversarial data. One approach to achieving structural robustness is by controlling the Lipschitz constant of the neural network, either through penalization (Gouk et al., 2021) or explicit architectural constraints (Anil et al., 2019). However, architectures that explicitly control the Lipschitz constant often impose significant constraints on model expressivity. While recent advances using learnable activation functions such as linear splines (Béthune et al., 2024) and GroupSort (Anil et al., 2019) have improved certified robustness, these approaches still face a fundamental trade-off: enforcing tight Lipschitz constraints limits the expressive capacity of the network, which in practice often results in degraded clean accuracy compared to unconstrained models (Anil et al., 2019; Tan & Wu, 2024). From this perspective, viewing neural networks as dynamical systems, particularly through Neural Ordinary

Differential Equations (Neural ODEs; (Chen et al., 2018)), allows researchers to apply stability concepts from Lyapunov theory when designing models. The key idea is that a robust classifier should act like a stable system: when an input is slightly perturbed from its original (nominal) value, the system's response should return to a stable state rather than drift away due to small perturbations (Kang et al., 2021; Rodriguez et al., 2022). Existing Lyapunov-based methods typically achieve robustness either by penalizing the vector field to encourage local stability around training samples (Rodriguez et al., 2022; Kang et al., 2021; Huang et al., 2024), or by constructing neural networks that inherently satisfy a Lyapunov stability criterion (Huang et al., 2022). However, the type of constraints imposed in these approaches only ensures that points within a neighborhood of the original data point converge to it, without specifying how large or small this neighborhood is. Furthermore, since these networks are typically trained using the cross-entropy loss, which is known not to promote the most robust solutions (Béthune et al., 2022), the resulting robustness improvements are often limited.

In this work, we take a different approach. Rather than imposing constraints on the learned vector field through penalties, we construct a vector field that explicitly admits asymptotically stable equilibrium points, thereby embedding Lyapunov stability directly into the model's dynamics. Moreover, the type of stability criterion we adopt provides additional insight into the region of convergence around each data point and allows us to formally show that, in an idealized setting, any point within the support of the input data distribution converges asymptotically to one of the stable equilibria. Notably, this dynamical behavior mirrors the concept of score estimation in score-based generative models (Song & Ermon, 2019; Song et al.): the induced flow guides points toward regions of high data density, revealing a principled connection between Lyapunov stability and score-based modeling that may foster mutual insights between these two domains (Song et al.; Lipman et al., 2023).

Our main contributions are: (1) a novel architecture embedding Lyapunov stability via vector fields with explicit asymptotically stable equilibria, (2) formal convergence analysis with characterized basins of attraction, (3) a principled connection to score-based generative modeling, and (4) empirical validation showing improved robustness over prior Lyapunov-based methods while maintaining competitive clean accuracy.

## 2 Background on Lyapunov Theory

Lyapunov theory enables the study of the stability of equilibrium points of the solutions of an ODE (Khalil, 2002). More specifically, it provides sufficient conditions for the simple (respectively asymptotic) stability of equilibrium points. Informally, an equilibrium point is stable if all solutions starting from nearby points stay nearby. Moreover, it is called asymptotically stable if all solutions starting from nearby points not only stay nearby but tend towards the equilibrium point. Finally, it is called exponentially stable equilibrium point if it is asymptotically stable and if we get nearby enough the solutions converge to it at least with an exponential rate.

Consider an autonomous dynamical system described by the equation:

$$\frac{d\mathbf{z}(t)}{dt} = f_\theta(\mathbf{z}(t)), \tag{1}$$

where $f_\theta \colon \mathbb{R} \times D \to \mathbb{R}^n$ a continuous locally Lipschitz function and $D \subseteq \mathbb{R}^n$. Its flow is denoted by:

$$\phi(t, \mathbf{z}(0)) = \mathbf{z}(0) + \int_0^t f_\theta(\mathbf{z}(t))\, dt\,.$$

*Remark.* This section discusses exclusively Lyapunov theory for autonomous dynamical systems. This formalism can be extended to non-autonomous dynamical systems, however, for the sake of simplicity, it will be omitted.

**Definition 2.1.** A point $\mathbf{z}_0$ is called an *equilibrium point* of the dynamical system $\dot{\mathbf{z}} = f_\theta(\mathbf{z})$ if

$$f_\theta(\mathbf{z}_0) = 0.$$

In this case, the associated flow satisfies

$$\phi(t, \mathbf{z}_0) = \mathbf{z}_0, \quad \forall t \in \mathbb{R}.$$

The equilibrium point $\mathbf{z}_0$ is said to be *stable* if for all $\varepsilon > 0$, there exists $\delta(\varepsilon) > 0$ such that

$$\|\mathbf{z}' - \mathbf{z}_0\| < \delta(\varepsilon) \implies \|\phi(t, \mathbf{z}') - \mathbf{z}_0\| < \varepsilon, \quad \forall t > 0.$$

It is called *asymptotically stable* if

$$\|\phi(t, \mathbf{z}') - \mathbf{z}_0\| \longrightarrow 0 \quad \text{as } t \to \infty.$$

Moreover, $\mathbf{z}_0$ is called *exponentially stable* if there exist constants $C, \gamma > 0$ such that

$$\|\phi(t, \mathbf{z}') - \mathbf{z}_0\| \leq Ce^{-\gamma t}\|\mathbf{z}' - \mathbf{z}_0\|, \quad \forall t > 0.$$

Conceptually, there is a natural connection between stability in dynamical systems and robustness in machine learning. In dynamical systems, stability captures the idea that the effect of a small perturbation should diminish over time. In adversarially robust classification, one seeks a similar behavior: predictions should not change significantly under small but worst-case perturbations of the input. This parallel has motivated the use of dynamical-systems-inspired tools in the design of robust learning architectures.

Neural ODEs provide a particularly convenient setting for this perspective, as they model the evolution of representations as continuous-time dynamics. From this viewpoint, robustness can be encouraged by shaping the vector field so that trajectories originating from perturbed inputs remain close to one another, or are driven back toward stable regions of the state space. Several recent works exploit this idea by explicitly enforcing stability properties of the induced dynamics while still carrying out a classification task.

These notions of stability can be made operational through simple sufficient conditions that allow one to assess or enforce the stability of equilibrium points. In practice, two main approaches are commonly used: **spectral characterization** and **Lyapunov function characterization**.

### 2.1 Spectral Characterization

The first type of characterization, which we refer to as spectral characterization, studies the eigenvalues of the Jacobian of the vector field (Khalil, 2002), which provide insight into the stability of a dynamical system. For a linear autonomous system:

$$\dot{\mathbf{z}}(t) = A\,\mathbf{z}(t), \tag{3}$$

the origin is stable if all eigenvalues of $A$ have negative real parts, and exponentially stable if they are strictly negative (see Appendix B.1). This result can be extended to nonlinear systems via the Hartman–Grobman theorem (Baratchart et al.), which states that near a hyperbolic equilibrium point, the nonlinear dynamics are topologically conjugate to their linearization. The full statement and proof are provided in Appendix B.2.

**SONet (Huang et al., 2022).** This approach constructs a provably Lyapunov-stable neural ODE-based classifier by designing the vector field with a skew-symmetric component and a damping term. The skew-symmetric structure enforces conservative dynamics, while the damping ensures that all eigenvalues of the Jacobian lie in the left half-plane, guaranteeing exponential stability. Further architectural and implementation details are provided in Appendix B.3.

**SODEF (Kang et al., 2021).** SODEF (Stable neural ODE with Lyapunov-stable equilibrium points) enforces Lyapunov stability through Jacobian regularization inspired by the Levy–Desplanques theorem, which links diagonal dominance to negative-real-part eigenvalues. Instead of constraining the architecture, SODEF introduces a loss term that encourages strict diagonal dominance of the Jacobian, thereby ensuring stability. This approach provides a trade-off between robustness and training efficiency. The formal statement of the Levy–Desplanques theorem and the exact regularization formulation are given in Appendix B.4.

**ASODE (Li et al., 2022b).** ASODE (A Stable neural ODE) extends the spectral stability framework to non-autonomous dynamics by introducing time-dependent vector fields. Unlike SODEF, which operates in the feature space, ASODE constructs a time-dependent neural ODE that acts as a denoising flow in the input space, transforming adversarial examples toward the clean data manifold. The denoised inputs are then fed to a standard classifier trained on clean data. By ensuring Lyapunov stability of the denoising flow through Jacobian regularization, ASODE achieves both certified and empirical robustness against adversarial perturbations.

## 2.2 Lyapunov Function Characterization

Another way to characterize equilibrium points is through *Lyapunov functions*, which generalize the concept of energy in physics (Yuan et al., 2010). Formally, they provide sufficient conditions for the stability of equilibrium points. Let $V : D \to \mathbb{R}$ be a continuously differentiable function on a domain $D \subseteq \mathbb{R}^n$ containing $\mathbf{z}_0$. If

$$V(\mathbf{z}_0) < V(\mathbf{z}), \ \forall \mathbf{z} \in D \setminus \{\mathbf{z}_0\}, \quad \text{and} \quad \dot{V} = \langle \nabla V, f(\mathbf{z}) \rangle \leq 0,$$

then $\mathbf{z}_0$ is a stable equilibrium point. If the inequality is strict, the equilibrium is asymptotically stable. If $V(\mathbf{z}(t)) \leq e^{-\kappa t} V(\mathbf{z}(0))$, it is exponentially stable (see Appendix C.1).

Learning such a Lyapunov function jointly with system dynamics is generally intractable due to the non-convex, multi-minima nature of the problem. Recent approaches thus focus on designing vector fields that *implicitly* satisfy Lyapunov conditions, ensuring stability by construction. Notable examples include LyaNet and Lyapunov-structured gradient systems.

**LyaNet** (Rodriguez et al., 2022) reinterprets the cross-entropy loss as a Lyapunov function $V_y(\mathbf{z})$ with a minimum at the correct label $y$. The neural ODE dynamics are constrained to ensure exponential decay of $V_y$ via a sufficient condition derived from Grönwall's inequality (Appendix C.3). This guarantees convergence toward the correct class while maintaining Lyapunov stability.

Another important family of systems are **gradient systems** (Khalil, 2002), where dynamics follow the steepest descent of a potential $V$:

$$\frac{d\mathbf{z}(t)}{dt} = -\nabla V(\mathbf{z}(t)).$$

Here, $V$ acts naturally as a Lyapunov function since $\dot{V} = -\|\nabla V\|^2 \leq 0$. Such systems inherently produce stable dynamics, with equilibrium points corresponding to the critical points of $V$. This structure has been leveraged in (Massaroli et al., 2020), where $V$ is parameterized by a neural network and the vector field is its gradient. Although this formulation ensures theoretical stability, its scalability to high-dimensional domains remains computationally challenging (see Appendix C.4).

To summarize, a growing body of work seeks to improve adversarial robustness by incorporating Lyapunov stability into neural ODE dynamics. These approaches provide an appealing theoretical framework and have led to encouraging empirical results. However, several open challenges remain, which limit the robustness guarantees that can be achieved in practice.

First, many existing methods, such as SODEF and LyaNet, operate in a latent feature space and therefore rely on an encoder that may itself be susceptible to adversarial perturbations. Second, stability is often promoted through soft regularization terms rather than enforced through hard architectural constraints, which may not ensure that the desired stability properties hold uniformly. Finally, the resulting Lyapunov conditions typically guarantee stability only in a local neighborhood around equilibrium points, whose size may be insufficient to account for adversarial perturbations of practical relevance.

Taken together, these observations highlight the need for alternative approaches to stability-based robustness that address these limitations. In particular, this motivates methods that (i) operate directly in the input space to avoid reliance on vulnerable encoders, (ii) enforce stability through architectural design rather than solely through soft regularization, and (iii) provide stability guarantees that extend beyond small local neighborhoods.

In the next section, we introduce a dynamical system framework that is designed with these objectives in mind.

## 3 Proposed Method

To address the limitations identified in the previous section, we propose a novel approach that leverages the gradient flow of the data probability density function as a provably stable dynamical system for adversarial robustness. Our method is built on three key ideas:

**(1) Score-based gradient flow:** We exploit the fact that the gradient of the log-density defines a dynamical system with asymptotically stable equilibria at the modes of the data distribution.

**(2) Input-space denoising:** By implementing this flow as a neural ODE operating directly in the input space, we create a denoiser that transforms adversarial examples toward clean modes without relying on vulnerable encoders.

**(3) Global convergence guarantees:** Under mild regularity conditions, we prove that all points in the support of the data distribution converge to stable equilibria, providing robustness beyond local neighborhoods.

This leads to a two-stage architecture (Figure 1): a neural ODE that implements the score-based gradient flow, followed by a standard classifier trained on clean data. Since stable equilibria are fixed points, clean inputs remain unchanged while adversarial perturbations are removed.

**Architecture design choices.** The neural ODE layer is parameterized by an integration time $t_f$, which controls the depth of the transformation and determines how far inputs are pushed along the gradient flow toward the nearest mode. Larger values of $t_f$ allow for stronger convergence but increase computational cost and sensitivity to score approximation errors.

Importantly, we train the score model and classifier independently. The score is learned using denoising score matching on the training data, while the classifier is trained on clean examples without exposure to the neural ODE transformation. This modular design ensures that the neural ODE layer remains reusable as a plug-and-play denoising component compatible with any off-the-shelf classifier, without requiring joint fine-tuning or architectural co-adaptation.

A critical property that makes this approach effective for classification is that the basins of attraction exhibit strong label consistency with their corresponding modes. When class-conditional distribution supports are disjoint, as empirical studies suggest for MNIST and CIFAR-10 (Yang et al., 2020), each basin corresponds to a single class. Even with overlapping supports, basins maintain significant semantic similarity with their equilibria.

We now formalize these ideas, beginning with the score function and its estimation, then analyzing the convergence properties of the resulting gradient system.

### 3.1 Score and its Estimation

**Definition (Score of a pdf (Hyvärinen, 2005)).** Let $p(\mathbf{x})$ be the probability density function of an absolutely continuous random variable $X$. The *score* of $p(\mathbf{x})$ is defined as the gradient of $\ln(p(\mathbf{x}))$ with respect to $\mathbf{x}$:

$$\nabla_{\mathbf{x}} \ln(p(\mathbf{x})).$$

The score is a central concept in *Langevin dynamics* (Teh et al., 2016; Hochberg et al., 2019; Langevin, 1908), which are stochastic differential equations used for sampling from complex distributions without computing normalizing constants. In generative modeling, score estimation underlies diffusion models, enabling training by matching a neural approximation $\psi_\theta(\mathbf{x})$ to the true score $\nabla_{\mathbf{x}} \ln(p(\mathbf{x}))$.

The classic *score matching* objective (Hyvärinen, 2005) minimizes the squared Euclidean distance between the estimated and true scores. After simplification and integration by parts (see Appendix D), this objective

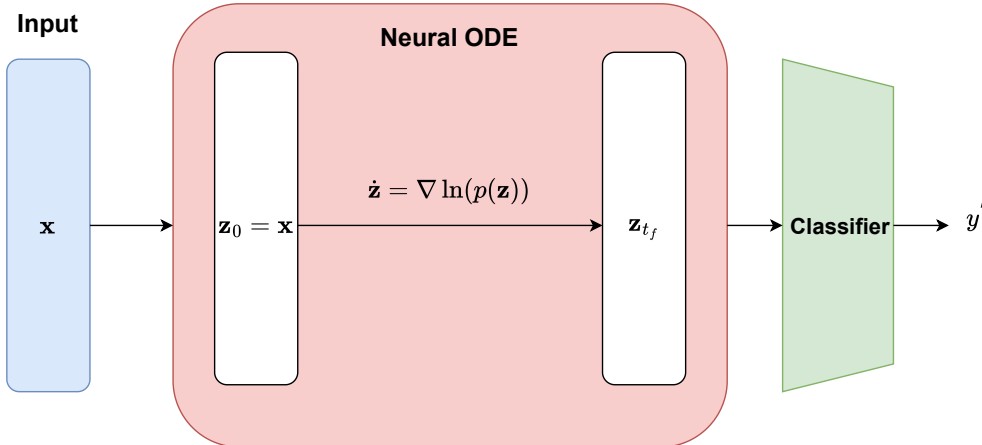

Figure 1: Neural ODE-based architecture for robust classification. The neural ODE layer implements the score gradient flow $d\mathbf{z}/dt = \nabla \ln p(\mathbf{z})$, driving inputs toward stable modes before classification.

can be expressed as:

$$\mathbb{E}_{p(\mathbf{x})}\left[\text{tr}(\nabla_{\mathbf{x}}\psi_\theta(\mathbf{x})) + \tfrac{1}{2}||\psi_\theta(\mathbf{x})||_2^2\right].$$

Because this involves computing the trace of the Jacobian, which scales quadratically with the data dimension, practical implementations use Hutchinson's stochastic trace estimator (Appendix D). Alternatively, (Vincent, 2011) proposed a denoising-based score matching approach that estimates the score of a Gaussian-perturbed distribution, leading to the modern noise-conditional score networks that form the basis of diffusion models (Song et al.).

### 3.2 Score as a Gradient System

Having established how to estimate the score $\nabla \ln p(\mathbf{z})$, we now show that the resulting gradient flow provides the stability guarantees required for robust classification.

A *gradient system* is a dynamical system where the state vector $\mathbf{z}(t)$ evolves according to the negative gradient of a scalar potential function:

$$\frac{d\mathbf{z}(t)}{dt} = -\nabla V(\mathbf{z}(t)).$$

If $V$ is continuously differentiable, bounded from below, and has isolated minima, these minima are asymptotically stable equilibrium points (Khalil, 2002).

Following this formulation, we define the *score gradient system* associated with the negative logarithm of a pdf:

$$\frac{d\mathbf{z}(t)}{dt} = \nabla \ln(p(\mathbf{z}(t))).$$

This system admits $-\ln(p(\mathbf{z}(t)))$ as a Lyapunov function, whose minima coincide with the modes of $p(\mathbf{z})$. Thus, under mild conditions, the dynamics drive any point in space toward one of these modes.

In contrast with stability-based models such as SODEF (Choi et al., 2021), ASODE (Li et al., 2022b), or SONet (Rodriguez et al., 2022), which enforce asymptotic stability locally via linearization, the score gradient system provides a global view of the attraction basins of the modes of the data distribution.

**Theorem 3.1** (Convergence to Limit Point (Lee et al., 2016)). *Assume that $V$ is continuously differentiable, has isolated critical points, and compact sublevel sets. Then for all $\mathbf{z}(0) \in \mathbb{R}^n$, $\lim_{t\to\infty} \mathbf{z}(t)$ exists and is a critical point of $V$.*

**Proposition 3.2** (Coercivity and Compactness (Burke)). *Let $h : \mathbb{R}^n \to \mathbb{R}$ be continuous on $\mathbb{R}^n$. The function $h$ is coercive if and only if, for every $\alpha \in \mathbb{R}$, the set $\{\mathbf{z} \mid h(\mathbf{z}) \le \alpha\}$ is compact.*

**Corollary 3.3.** *The sublevel sets of* $-\ln(p(\mathbf{z}))$ *are compact (proof in Appendix E).*

This corollary guarantees that trajectories under the score gradient system converge to finite limit points, corresponding to the modes of $p(\mathbf{z})$. The discrete version of this flow,

$$\mathbf{z}_{k+1} = \mathbf{z}_k + \tau \nabla \ln(p(\mathbf{z}_k)),$$

is reminiscent of Langevin dynamics without noise, driving data points deterministically toward high-density regions. This process effectively partitions the input space into basins of attraction centered on each mode, providing a natural geometric interpretation of the learned distribution.

Further details on numerical stability and Lyapunov-preserving integrators are discussed in Appendix E.2.

**Complete pipeline.** Our method trains independently (1) a score model $\psi_\theta$ via denoising score matching and (2) a classifier $g$ on clean data. At inference, given input $\mathbf{x}_0$, we denote the neural ODE state at time $t$ as $\mathbf{z}(t)$ with $\mathbf{z}(0) = \mathbf{x}_0$, solve $d\mathbf{z}/dt = \psi_\theta(\mathbf{z})$ to obtain $\mathbf{z}(t_f)$, then classify via $g(\mathbf{z}(t_f))$. This design ensures that all points, clean instances or their adversarially perturbed counterparts, converge to the nearest mode.

## 4 Experiments

We conduct a comprehensive experimental evaluation to assess the effectiveness of the proposed method. We first provide intuition using a synthetic two-dimensional dataset where the score can be computed analytically, allowing us to study the influence of key hyperparameters and compare learned approximations to ground truth. We then evaluate performance on real-world image classification benchmarks and compare against state-of-the-art Lyapunov-based methods for adversarial robustness.

### 4.1 Experimental Setup

**Datasets.** We evaluate on one synthetic dataset and four image classification benchmarks.

*Two coiling spirals:* A synthetic two-dimensional dataset where each class follows a Gaussian distribution arranged in a spiral pattern. This dataset serves two purposes: (1) analytical computation of the true score, enabling direct comparison between exact and learned vector fields, and (2) a known Bayes-optimal classifier, providing a theoretical upper bound for comparison.

*MNIST (Fawzi et al., 2016; Wong & Kolter, 2018):* A dataset of 28×28 grayscale handwritten digits (10 classes, 60K training, 10K test images). MNIST is widely used in adversarial robustness literature and exhibits near-disjoint class supports (Yang et al., 2020), ideal for validating our theoretical predictions.

*CIFAR-10 (Cohen et al., 2019):* A dataset of 32×32 color natural images (10 classes, 50K training, 10K test images). CIFAR-10 provides a more challenging testbed with higher-dimensional inputs and more complex visual patterns.

*DermaMNIST and PneumoniaMNIST (Yang et al., 2023):* Two medical imaging datasets from the MedMNIST collection. DermaMNIST contains dermatoscopic images for skin lesion classification (7 classes), while PneumoniaMNIST contains chest X-rays for pneumonia detection (2 classes). These datasets are chosen to assess robustness in domain-specific applications where class boundaries may overlap due to diagnostic ambiguity, and to evaluate generalization beyond natural images.

Additional dataset statistics and preprocessing details are provided in Appendix F.

**Architectural Details.** We compare our approach against four Lyapunov-based baselines: LyaNet (Rodriguez et al., 2022), SODEF (Kang et al., 2021), ASODE (Li et al., 2022a), and SONet (Huang et al., 2022). To ensure fair comparison, all methods use the same classifier architectures: ResNet-18 for MNIST, the standard RobustBench model (Croce et al.) for CIFAR-10, and the architectures recommended in (Yang et al., 2023) for medical datasets.

For our method, we evaluate two variants of the score network: (1) a three-layer MLP with tanh activations, matching the computational complexity of ASODE's and SODEF's vector fields, and (2) state-of-the-art

diffusion models, DDPM (Ho et al., 2020) and NCSN++ (Song et al.), to demonstrate scalability to high-capacity architectures.

For the two-dimensional spirals dataset, we derive an analytical expression for the score, enabling exact computation of the gradient flow (details in Appendix F).

**ODE Solver Configuration.** Neural ODE integration uses the `dopri5` adaptive solver with $t_f = 1.0$, `rtol`$=10^{-4}$, and `atol`$=10^{-4}$. We backpropagate through the stored computational graph rather than using the adjoint method, enabling exact gradients for adversarial attack generation. Experiments use batch size 50 on an NVIDIA Tesla T4 GPU.

**Adversarial Attacks.** We evaluate robustness using two white-box attacks following the protocols of (Li et al., 2022c; Kang et al., 2021):

*Fast Gradient Sign Method (FGSM) (Goodfellow et al., 2015):* A single-step attack that perturbs inputs in the direction of the gradient of the loss with respect to the input: $\mathbf{x}_{adv} = \mathbf{x} + \epsilon \cdot \text{sign}(\nabla_{\mathbf{x}} \mathcal{L})$.

*Projected Gradient Descent (PGD) (Madry et al.):* An iterative variant of FGSM that repeatedly applies small perturbations and projects back onto the $\ell_\infty$ ball. We use 20 iterations with step size $\alpha = \epsilon/4$.

We use perturbation budgets of $\epsilon = 0.3$ for MNIST (standard for grayscale images) and $\epsilon = 8/255$ for CIFAR-10, DermaMNIST, and PneumoniaMNIST (standard for RGB images). These values are widely adopted in the adversarial robustness literature (Madry et al.; Carlini & Wagner, 2017).

We report two metrics: *clean accuracy* (performance on unperturbed test data) and *robust accuracy* (performance on adversarially perturbed data). Further implementation details are in Appendix G.

## 4.2 Two-Dimensional Dataset

We analyze the behavior of our method on the two coiling spirals dataset, where each class follows a Gaussian distribution. This dataset offers unique advantages: (1) the score can be computed analytically, enabling comparison between exact and learned vector fields, (2) the Bayes-optimal classifier is known in closed form, providing a theoretical benchmark with provable robustness properties (Richardson & Weiss, 2021), and (3) the Gaussian structure allows us to study the effect of noise injection during score estimation.

**Dataset properties and hyperparameter selection.** The two classes are generated using Gaussian distributions with equal covariances. As described in Section 3.1, score estimation can be performed on noise-perturbed versions of the data with noise level $\sigma$. For Gaussian data, adding Gaussian noise preserves the modes while increasing covariances, meaning the score field points in the same direction for all $\sigma$ values below a threshold $\sigma'$, beyond which the distribution collapses into a single mode. For this dataset, we use $\sigma = 0$ (the exact data distribution) to study the baseline behavior of our method.

**Decision boundary evolution.** Figure 2 shows the decision boundaries of the MLP classifier (left) and the Bayes-optimal Gaussian classifier (right). While the MLP achieves high accuracy (98.90%), its decision boundary exhibits irregularities in low-density regions far from the training data.

Figure 3 shows how the decision boundary of our complete model (neural ODE + MLP classifier) evolves as the integration time $t_f$ increases. At $t_f = 0$, the neural ODE acts as the identity, so the boundary matches the MLP. As $t_f$ increases, inputs are progressively pushed toward their nearest mode. At $t_f \to \infty$, each point converges to its closest mode, and the decision boundary reflects the label of each mode expanded across its entire basin of attraction, which for this Gaussian case with disjoint supports, corresponds exactly to the Bayes-optimal classifier.

**Effect of integration time on accuracy.** Table 1 quantifies this effect using the exact analytical score. As $t_f$ increases, accuracy improves monotonically from 98.90% (matching the MLP at $t_f = 0$) to 99.70% at $t_f = 5$, asymptotically approaching the Bayes-optimal accuracy of 99.75%.

**Exact vs. approximated score.** In practice, the score must be learned from data rather than computed analytically. Table 2 compares performance using the exact score versus a three-layer MLP with tanh activations trained using the score matching objective (Eq. 3.1).

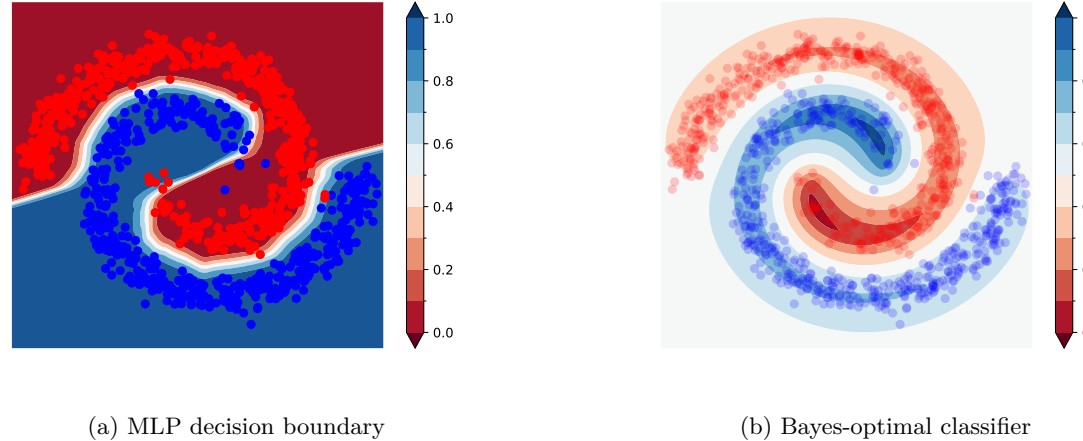

(a) MLP decision boundary                                    (b) Bayes-optimal classifier

Figure 2: Decision boundaries for the MLP (left) and the Bayes-optimal Gaussian classifier (right). The MLP achieves 98.90% accuracy but produces irregular boundaries in low-density regions.

Decision boundary for various values of integration time $t_f$

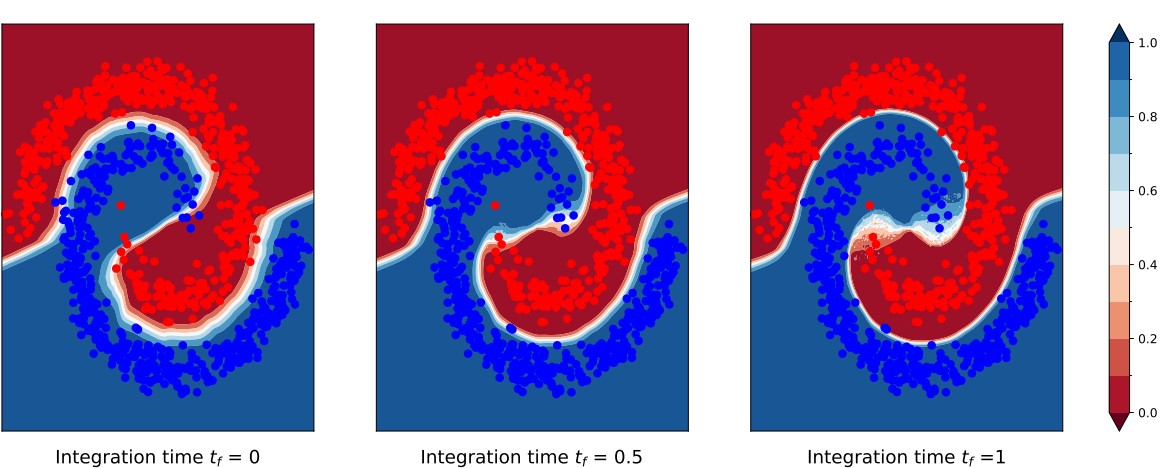

Integration time $t_f = 0$          Integration time $t_f = 0.5$          Integration time $t_f = 1$

Figure 3: Evolution of the decision boundary as integration time $t_f$ increases. The boundary transitions from the MLP's irregular surface (at $t_f = 0$) toward the Bayes-optimal boundary (as $t_f \to \infty$).

For small $t_f$, the approximation closely matches the exact score, yielding nearly identical accuracy. However, at $t_f = 5$, the approximation error accumulates over the longer integration, causing a drop from 99.70% to 99.40%.

This reveals two practical challenges for choosing $t_f$: (1) computational cost increases with $t_f$, and (2) when the score is only an approximation, large integration times can cause trajectories to deviate from the true modes. These results suggest that moderately large values (e.g., $t_f = 1$) provide the best trade-off between convergence to modes, computational efficiency, and robustness to approximation errors. We use $t_f = 1$ for all subsequent experiments.

### 4.3   Results on Image Datasets

We now evaluate our method on real-world image classification tasks and compare against Lyapunov-based baselines.

| Method | Accuracy (%) |
|---|---|
| MLP | 98.90 |
| Ours ($t_f = 0$) | 98.90 |
| Ours ($t_f = 0.5$) | 99.23 |
| Ours ($t_f = 1$) | 99.50 |
| Ours ($t_f = 5$) | 99.70 |
| Bayes-optimal | 99.75 |

Table 1: Accuracy as a function of integration time $t_f$ on the two coiling spirals dataset (exact score). Performance improves monotonically and approaches the Bayes-optimal bound.

| Integration time | Exact score (%) | Approximated score (%) |
|---|---|---|
| $t_f = 0$ | 98.90 | 98.90 |
| $t_f = 0.5$ | 99.23 | 99.20 |
| $t_f = 1$ | 99.50 | 99.48 |
| $t_f = 5$ | 99.70 | 99.40 |

Table 2: Comparison of exact vs. learned score on the two coiling spirals dataset. The learned score closely matches the exact score for moderate $t_f$, but accumulates error for large $t_f$.

**Effect of noise level $\sigma$.** As described in Section 3.1, the noise level $\sigma$ used during score training determines which perturbed density we approximate. When $\sigma = 0$, we match the original data density exactly. For datasets with disjoint class supports (MNIST, CIFAR-10), this should align with the Bayes-optimal decision boundary, as demonstrated on the synthetic dataset. For medical datasets where diagnostic ambiguity may create overlapping class supports, the optimal $\sigma$ is less clear a priori.

We parameterize the noise level as $\sigma(l)$ where $l \in [0, 1]$ and $\sigma(l)$ increases monotonically with $l$. We perform a sweep over $l$ and select the value maximizing robust accuracy on a validation set (detailed results in Appendix Figure 8). Empirically, we find that robustness tends to increase as $l \to 0$ (i.e., $\sigma \to 0$) across all datasets, suggesting substantial class separation even in medical imaging. At $l = 1$ (maximum noise level), the score field becomes very weak and the neural ODE layer has minimal effect.

**Qualitative analysis.** Adversarial perturbations degrade model predictions while preserving semantic content from a human perspective. Appendix Figures 7 and 9 show examples of adversarial images before and after the neural ODE transformation. The transformation tends to reduce perturbations, with visible corrections occurring primarily near object boundaries and high-frequency regions where adversarial noise is most concentrated.

**Quantitative comparison.** Table 3 reports clean and robust accuracy across all methods and datasets. Our method with diffusion-based score networks (NCSN++) achieves the strongest robust accuracy across all datasets. On MNIST, NCSN++ achieves 97.39% FGSM accuracy and 96.50% PGD accuracy, compared to 75.12% and 56.33% for the best baseline (SONet), improvements of over 22 and 40 percentage points respectively. On CIFAR-10, NCSN++ achieves 87.00% FGSM accuracy and 82.57% PGD accuracy, versus 73.44% and 62.35% for SONet, gains of approximately 14 and 20 percentage points. On medical datasets, NCSN++ achieves 75.39% FGSM and 74.02% PGD accuracy on DermaMNIST, and 82.81% FGSM and 73.14% PGD accuracy on PneumoniaMNIST, outperforming all baselines.

The simpler MLP-based score variant shows mixed results: it performs well on MNIST (92.23% FGSM, 92.00% PGD) and achieves competitive results on medical datasets (69.39%/62.00% on DermaMNIST, 82.00%/72.20% on PneumoniaMNIST), but underperforms on CIFAR-10 (67.40% FGSM, 47.50% PGD) compared to several baselines. This suggests that score approximation quality is critical for robustness on complex natural image datasets.

Clean accuracy is slightly lower than some baselines (e.g., ASODE on CIFAR-10: 95.16% vs. our 90.62%), likely because the neural ODE introduces small deformations even for clean inputs. However, this modest

Table 3: Clean and robust accuracy (%) across datasets under FGSM and PGD attacks. "−" indicates unavailable results due to missing or non-reproducible code. Best results in **bold**.

| Method | Dataset | Clean | FGSM | PGD |
|---|---|---|---|---|
| SODEF | MNIST | 99.44 | 63.36 | 45.25 |
| | CIFAR-10 | 95.00 | 68.05 | 55.59 |
| | DermaMNIST | – | – | – |
| | PneumoniaMNIST | – | – | – |
| ASODE | MNIST | 99.44 | 65.13 | 46.85 |
| | CIFAR-10 | 95.16 | 69.94 | 57.35 |
| | DermaMNIST | – | – | – |
| | PneumoniaMNIST | – | – | – |
| LyaNet | MNIST | 99.34 | 47.96 | 47.11 |
| | CIFAR-10 | 82.70 | 57.62 | 41.47 |
| | DermaMNIST | 68.08 | 61.48 | 51.00 |
| | PneumoniaMNIST | 88.14 | 79.65 | 66.03 |
| SONet | MNIST | 99.27 | 75.12 | 56.33 |
| | CIFAR-10 | 91.50 | 73.44 | 62.35 |
| | DermaMNIST | 70.62 | 62.16 | 53.54 |
| | PneumoniaMNIST | 88.69 | 79.55 | 67.75 |
| Ours (MLP) | MNIST | 97.20 | 92.23 | 92.00 |
| | CIFAR-10 | 79.40 | 67.40 | 47.50 |
| | DermaMNIST | 73.66 | 69.39 | 62.00 |
| | PneumoniaMNIST | 87.00 | 82.00 | 72.20 |
| Ours (NCSN++) | MNIST | 98.11 | **97.39** | **96.50** |
| | CIFAR-10 | 90.62 | **87.00** | **82.57** |
| | DermaMNIST | **75.58** | **75.39** | **74.02** |
| | PneumoniaMNIST | 86.14 | **82.81** | **73.14** |

trade-off (typically 2-5 percentage points) is acceptable given the substantial robustness gains, over 25 percentage points improvement in PGD accuracy on CIFAR-10 compared to ASODE (82.57% vs. 57.35%).

## 5 Discussion

**Robustness to Adversarial Attacks.**
Achieving robustness against adversarial attacks remains an active area of research within the deep learning community. A wide range of methods have been proposed to address this challenge, including modifications to network architectures, loss functions, and training procedures. One particularly promising line of work views neural networks through the lens of dynamical systems, as introduced by neuralODEs. This connection has enabled the application of tools from Lyapunov stability theory to guide the design of models that are inherently more robust to adversarial perturbations.

The method we propose builds directly on this perspective. It leverages Lyapunov theory to enhance robustness, with a novel contribution in decoupling the Lyapunov-stable neuralODE layer from the classifier layers. This modular design enables the purification of adversarial inputs independently from classification. Empirical evaluations across several datasets show that our approach consistently outperforms state-of-the-art defenses under two types of adversarial attacks. Furthermore, qualitative analysis suggests that the Lyapunov-stable layer acts as a denoising module, providing both robustness and interpretability. At a deeper level, our work highlights a connection between score estimation and Lyapunov stability, which opens new avenues for integrating dynamical systems theory into adversarial defense strategies.

**Relation to Adversarial Purification.**
Our approach aligns with the broader family of adversarial purification methods, which aim to map adversarial inputs back to regions of high likelihood under the data distribution. Specifically, our method employs score estimation to enhance the robustness of a pretrained classifier without requiring its retraining. This is achieved through a dynamical systems framework guided by Lyapunov stability.

The idea is conceptually related to early purification-based defenses, such as PixelDefend (Song et al., 2018), which observed that adversarial examples tend to have low likelihood under a density estimator trained on clean data. Purification methods attempt to transform such perturbed inputs into more probable counterparts that are likely to be correctly classified. Later approaches, such as (Yoon et al., 2021), utilize advances in score-based modeling, applying the score from the NCSN architecture to iteratively increase the input's likelihood. Their method removes stochasticity from Langevin dynamics, resulting in a deterministic transformation reminiscent of Euler discretization, similar in form to our proposed ODE-based method. However, their approach has key limitations: it is heuristic and lacks theoretical guarantees, and it is memory-intensive as it requires storing the entire computation graph. In contrast, our method employs an ODE solver and leverages the adjoint method, enabling memory-efficient training.

Another closely related line of work is (Nie et al., 2022), which uses the continuous-time formulation of stochastic differential equations (SDEs) for purification. In this framework, adversarial inputs are injected with increasing Gaussian noise, a process known as the backward phase of the diffusion process, until the input nears the data manifold without losing semantic content. This is followed by a reverse diffusion step that brings the input back to the data distribution. Their approach benefits from established techniques in the SDE literature, including adjoint-based gradient computation for adversarial robustness.

**Limitations and Future Work.**
While our approach demonstrates improved adversarial robustness, it introduces a slight reduction in clean accuracy due to the decoupling of the denoising and classification modules. One possible direction to mitigate this drawback is to incorporate classification-relevant information during the training of the score function, for instance, via an auxiliary penalty term. Another limitation concerns the dependence of the method on hyperparameter tuning, which currently lacks a principled or automated selection strategy. This challenge is further exacerbated by the approximate nature of the learned score function, which may not precisely correspond to the gradient of a convergent flow or a scalar potential.

Additionally, our method currently lacks formal robustness guarantees. Integrating robustness certificates, such as those derived from Lyapunov stability theory, could provide stronger theoretical underpinnings. Previous works have proposed Lyapunov-inspired neural architectures that offer provable stability within a bounded perturbation radius (Rodriguez et al., 2022).

Beyond these considerations, a key insight from our work is the connection established between Lyapunov stability theory and score estimation. In our approach, this relationship was used to inform the choice of a dynamical system that benefits from score-driven guidance. Looking forward, this connection may also be exploited in the reverse direction, that is, to improve the quality of score estimation using tools from Lyapunov theory. In particular, while score models are ideally expected to represent gradients of scalar functions, contemporary architectures often relax this constraint in favor of increased expressiveness and scalability, focusing instead on minimizing the score-matching loss. These models could potentially benefit from regularization strategies inspired by the Variable Gradient Method from Lyapunov theory, encouraging closer adherence to true gradient behavior. Furthermore, exploring the integration of classification objectives into the score learning process may offer a path toward improving both robustness and clean accuracy.

# 6   Conclusion

In this article, we have presented a novel method for building robust neural networks against adversarial attacks. In particular, we leverage score estimation from the generative modeling community and dynamical system perspective of neural networks along with the Lyapunov notion of stability to devise a novel architecture with robustness properties. We showed that our methods outperforms the state of the art of the lyapunov based architectures when it comes to robustness with slight drop compared to the clean accuracy. By integrating Lyapunov theory with score models and adversarial robustness, we offer a fresh perspective on density estimation, robust classification, and adversarial purification. This approach opens up new avenues for robustifying neural networks and improving the performance of score models.

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

# A   Appendix

*Proof of Theorem 2.1.* **Proof:** The solution to the system can be written as:

$$\mathbf{z}(t) = \exp(At)\,\mathbf{z}(0).$$

$A$ can also be written as $PJP^{-1}$, where $P \in \mathbb{R}^{n \times n}$, and $J$ is the Jordan normal form, i.e., a block-diagonal matrix ($J = \mathrm{diag}(J_1, \cdots, J_r)$):

$$J_k = \begin{bmatrix} \lambda_k & 1 & 0 & \cdots & \cdots & 0 \\ 0 & \lambda_k & 1 & 0 & \cdots & 0 \\ \vdots & & \ddots & & & \vdots \\ \vdots & & & & \ddots & \vdots \\ \vdots & & & & \ddots & 1 \\ 0 & \cdots & \cdots & \cdots & 0 & \lambda_k \end{bmatrix} \in \mathbb{R}^{m_k \times m_k}. \tag{4}$$

We have $e^{Jt} = \mathrm{diag}(e^{J_1 t}, \cdots, e^{J_r t})$, where $J_k = \mathrm{diag}(\lambda_k) + N_k$, with $N_k$ a nilpotent matrix. Then:

$$e^{J_k t} = e^{\lambda_k t} e^{N_k t}.$$

As $N_k$ is nilpotent, and by expanding $e^{N_k t}$ as a power series, we get:

$$e^{N_k t} = \begin{bmatrix} 1 & t & \frac{t^2}{2} & \cdots & \cdots & \frac{t^{m_k-1}}{(m_k-1)!} \\ 0 & 1 & t & \frac{t^2}{2} & \cdots & \frac{t^{m_k-2}}{(m_k-2)!} \\ \vdots & & \ddots & & & \vdots \\ \vdots & & & & \ddots & t \\ 0 & \cdots & \cdots & \cdots & 0 & 1 \end{bmatrix}. \tag{5}$$

It can be verified that when the real part of $\lambda_k$ is strictly less than zero, $\|e^{J_k t}\| \to 0$ as $t \to +\infty$. If every $\lambda_k$ satisfies this condition, then the system is exponentially stable. If even one $\lambda_k$ has a real part strictly greater than zero, then the system is unstable. Finally, if one $\lambda_k$ has a real part equal to zero, then in order for the system to be stable, the polynomial term in $e^{N_k t}$ must vanish. This corresponds to $m_k = 1$, where $m_k$ is the geometric multiplicity of $\lambda_k$.

## B    Additional Theoretical and Architectural Details

### B.1    Spectral Characterization

**Theorem B.1** (Spectral Stability of Linear Systems)**.** *Consider the linear autonomous system*

$$\dot{\mathbf{z}}(t) = A\,\mathbf{z}(t), \tag{6}$$

*where $A \in \mathbb{R}^{n \times n}$.*

- *The origin is **stable** if and only if all eigenvalues of $A$ have non-positive real parts, and those with purely imaginary parts are simple roots of the minimal polynomial of $A$.*

- *The origin is **exponentially stable** if and only if all eigenvalues of $A$ have strictly negative real parts.*

*Proof.* Let $A = PJP^{-1}$ be the Jordan decomposition of $A$, with $J$ composed of Jordan blocks associated with eigenvalues $\lambda_i$. The general solution is $\mathbf{z}(t) = e^{At}\mathbf{z}(0) = Pe^{Jt}P^{-1}\mathbf{z}(0)$. Stability depends on $\|e^{At}\|$: if all $\Re(\lambda_i) < 0$, the exponential term decays exponentially, yielding exponential stability. If some eigenvalues lie on the imaginary axis but are simple, the oscillatory terms remain bounded, ensuring stability but not exponential decay. $\qquad \square$

## B.2 Hartman–Grobman Theorem

**Theorem B.2** (Hartman–Grobman (Baratchart et al.))**.** *Let* $\mathbf{z}_0$ *be a hyperbolic equilibrium point of the nonlinear system*

$$\dot{\mathbf{z}}(t) = f_\theta(\mathbf{z}(t)),$$

*i.e., the Jacobian* $\nabla_{\mathbf{z}_0} f_\theta$ *has no eigenvalue with zero real part. Then, there exists a neighborhood* $\mathcal{V}$ *of* $\mathbf{z}_0$*, a neighborhood* $\mathcal{O}$ *of 0, and a homeomorphism* $h : \mathcal{V} \to \mathcal{O}$ *such that the flow* $\phi$ *of* $f_\theta$ *satisfies*

$$h(\phi(t, \mathbf{z})) = e^{t\nabla_{\mathbf{z}_0} f_\theta} h(\mathbf{z}), \quad \forall \mathbf{z} \in \mathcal{V}, \ t \in \mathbb{R}.$$

*Remark.* The theorem implies that near a hyperbolic equilibrium point, the nonlinear system is topologically conjugate to its linearization. Consequently, local stability properties of nonlinear systems can be inferred from the spectral properties of their Jacobian.

## B.3 SONet Architecture Details

The SONet model (Huang et al., 2022) constructs a provably Lyapunov-stable neural ODE classifier by designing the vector field as:

$$f_\theta(\mathbf{z}) = \sigma\left(\begin{bmatrix} 0 & \mathbf{W} \\ -\mathbf{W}^\top & 0 \end{bmatrix} \mathbf{z} - \gamma \mathbf{I} \mathbf{z}\right), \tag{7}$$

where $\mathbf{W}$ is a trainable matrix, $\sigma$ is a strictly increasing activation function, and $\gamma > 0$ is a damping factor.

The skew-symmetric component ensures conservative, energy-preserving dynamics, while the damping term $-\gamma \mathbf{I}$ introduces controlled dissipation. The resulting Jacobian has eigenvalues with strictly negative real parts, ensuring exponential stability. The overall architecture integrates this stable ODE block with a linear classification layer. Although stability is guaranteed, classification relies on the final linear layer rather than on asymptotic convergence to class-specific attractors.

## B.4 SODEF Regularization Formulation

SODEF (Kang et al., 2021) enforces Lyapunov stability by imposing structural constraints on the Jacobian, inspired by the Levy–Desplanques theorem.

**Theorem B.3** (Levy–Desplanques Theorem)**.** *If* $A \in \mathbb{C}^{n \times n}$ *is strictly diagonally dominant with real and negative diagonal entries, i.e.*

$$|a_{ii}| > \sum_{j \neq i} |a_{ij}|, \quad a_{ii} < 0, \ \forall i,$$

*then* $A$ *is nonsingular and all its eigenvalues have negative real parts.*

Instead of computing eigenvalues explicitly, SODEF introduces a regularization loss that promotes strict diagonal dominance of the Jacobian:

$$\alpha_1 \|f_\theta(\mathbf{z}(0))\|^2 + \alpha_2 \, g_1\left(\sum_{i=1}^{n} [\nabla f_\theta(\mathbf{z}(0))]_{ii}\right)$$

$$+ \alpha_3 \, g_2\left(\sum_{i=1}^{n}\left(-\left|[\nabla f_\theta(\mathbf{z}(0))]_{ii}\right| + \sum_{j \neq i}\left|[\nabla f_\theta(\mathbf{z}(0))]_{ij}\right|\right)\right), \tag{8}$$

where $\alpha_1$, $\alpha_2$, and $\alpha_3$ are weighting coefficients, and $g_1$, $g_2$ are monotonically increasing functions. This regularization indirectly enforces stability while remaining compatible with standard neural network architectures. Although this approach may slow convergence, it provides a practical balance between robustness and expressivity.

$\square$

# C   Additional Material on Lyapunov Characterization

## C.1   Lyapunov Function Definition and Stability Conditions

Lyapunov functions provide sufficient conditions for the stability of equilibrium points in nonlinear systems.

**Definition (Lyapunov Function).** Let $D \subseteq \mathbb{R}^n$ be a domain containing $\mathbf{z}_0$. A continuously differentiable function $V : D \to \mathbb{R}$ is called a *Lyapunov function* for the system $\dot{\mathbf{z}} = f(\mathbf{z})$ if:

- $V(\mathbf{z}_0) < V(\mathbf{z})$ for all $\mathbf{z} \in D \setminus \{\mathbf{z}_0\}$,

- $\dot{V} = \langle \nabla V, f(\mathbf{z}) \rangle \leq 0$.

Then $\mathbf{z}_0$ is a stable equilibrium point. If $\dot{V} < 0$, $\mathbf{z}_0$ is *asymptotically stable*. If there exists $\kappa > 0$ such that

$$V(\mathbf{z}(t)) \leq e^{-\kappa t} V(\mathbf{z}(0)),$$

then $\mathbf{z}_0$ is *exponentially stable*. These notions of stability are local, holding within a neighborhood of $\mathbf{z}_0$ where $V$ decreases along trajectories.

## C.2   Grönwall's Inequality

Grönwall's inequality is a key tool for establishing exponential decay in Lyapunov-based proofs.

**Lemma (Grönwall's Inequality).** Let $u, \beta : [a, \infty) \to \mathbb{R}$. If

$$u'(t) \leq \beta(t) u(t),$$

then

$$u(t) \leq u(a) \exp \left( \int_a^t \beta(s) \, ds \right).$$

If $\beta(t) = -\kappa < 0$, we obtain

$$u(t) \leq u(a) e^{-\kappa(t-a)}.$$

This relation provides a sufficient condition for exponential stability when $u(t)$ represents a Lyapunov function along system trajectories.

## C.3   LyaNet: Exponentially Stable Neural ODEs

LyaNet (Rodriguez et al., 2022) reinterprets the cross-entropy loss as a Lyapunov function $V_y(\mathbf{z})$ minimized at the true class $y$. The neural ODE block enforces the exponential stability condition:

$$\frac{d}{dt} V_y(\mathbf{z}(t)) \leq -\kappa V_y(\mathbf{z}(t)),$$

which ensures that trajectories converge exponentially to the correct class. This condition can be equivalently written as:

$$\langle \nabla V_y(\mathbf{z}(t)), f_\theta(\mathbf{z}(t)) \rangle \leq -\kappa V_y(\mathbf{z}(t)).$$

During training, violations of this inequality are penalized through the loss:

$$\mathcal{L}_{\text{stab}} = \max \left\{ 0, \, \langle \nabla V_y(\mathbf{z}(t)), f_\theta(\mathbf{z}(t)) \rangle + \kappa V_y(\mathbf{z}(t)) \right\}.$$

This Lyapunov-based regularization ensures stable convergence of trajectories while maintaining discriminative power through the classification loss.

### C.4 Gradient Systems and Lyapunov Stability

Gradient systems represent a fundamental class of stable dynamical systems.

**Definition (Gradient System).** A system is said to be a *gradient system* if

$$\frac{d\mathbf{z}(t)}{dt} = -\nabla V(\mathbf{z}(t)),$$

for some scalar potential $V : \mathbb{R}^n \to \mathbb{R}$. In this case, $V$ naturally acts as a Lyapunov function since:

$$\dot{V}(\mathbf{z}) = \langle \nabla V, -\nabla V \rangle = -\|\nabla V\|^2 \leq 0,$$

with equality if and only if $\mathbf{z}$ is a critical point of $V$.

In (Massaroli et al., 2020), $V$ is parameterized by a neural network, and the vector field $f_\theta(\mathbf{z}) = -\nabla_{\mathbf{z}} V_\theta(\mathbf{z})$ is obtained via automatic differentiation. This construction ensures that the dynamics are Lyapunov-stable by design. However, extending this formulation to high-dimensional spaces (e.g., images) is computationally demanding due to the cost of computing gradients through deep architectures.

## D Additional Material on Score Estimation

### D.1 Derivation of the Score Matching Objective

Starting from the original score matching objective (Hyvärinen, 2005):

$$\mathbb{E}_{p(\mathbf{x})} \|\nabla_{\mathbf{x}} \ln(p(\mathbf{x})) - \psi_\theta(\mathbf{x})\|_2^2,$$

expanding the square and removing the constant term independent of $\theta$ yields:

$$\mathbb{E}_{p(\mathbf{x})} \left[ -2\nabla_{\mathbf{x}} \ln(p(\mathbf{x}))^\top \psi_\theta(\mathbf{x}) + \|\psi_\theta(\mathbf{x})\|_2^2 \right].$$

Using integration by parts, the first term becomes an expectation involving the divergence of $\psi_\theta$. This leads to:

$$\mathbb{E}_{p(\mathbf{x})} \left[ \mathrm{tr}(\nabla_{\mathbf{x}} \psi_\theta(\mathbf{x})) + \tfrac{1}{2} \|\psi_\theta(\mathbf{x})\|_2^2 \right].$$

This objective is equivalent to the original one but avoids explicit dependence on $\nabla \ln(p(\mathbf{x}))$.

### D.2 Hutchinson's Trace Estimator

The trace term $\mathrm{tr}(\nabla_{\mathbf{x}} \psi_\theta(\mathbf{x}))$ requires computing all partial derivatives, which scales as $\mathcal{O}(n^2)$. Hutchinson's trick approximates this using a random vector $\mathbf{v} \sim \mathcal{N}(0, I)$:

$$\mathrm{tr}(J_\psi) = \mathbb{E}_{\mathbf{v}}[\mathbf{v}^\top J_\psi \mathbf{v}].$$

This reduces the complexity to a single Jacobian-vector product, computable efficiently with automatic differentiation.

### D.3 Denoising-Based Score Estimation

Vincent (Vincent, 2011) proposed injecting Gaussian noise $\mathbf{n} \sim \mathcal{N}(0, \sigma^2 I)$ to form a perturbed density $p_\sigma(\mathbf{x})$, and training $\psi_\theta$ to approximate $\nabla_{\mathbf{x}} \ln(p_\sigma(\mathbf{x}))$. This avoids computing Jacobians directly and forms the foundation of noise-conditional score networks.

## E Proofs and Technical Results for the Score Gradient System

### E.1 Proof of Corollary on Compactness

Since $p(\mathbf{z}) \to 0$ as $\|\mathbf{z}\| \to +\infty$, we have $-\ln(p(\mathbf{z})) \to +\infty$. Therefore, $-\ln(p(\mathbf{z}))$ is coercive. By Proposition E, coercivity implies compact sublevel sets, completing the proof.

### E.2 Numerical Integration and Lyapunov Preservation

In practice, continuous dynamics $\dot{\mathbf{z}} = f(\mathbf{z})$ are approximated by discrete updates. While the continuous system ensures monotonic decrease of $V(\mathbf{z}(t))$, the discrete version may not preserve this property unless specific integrators are used.

Two classes of methods are known to preserve Lyapunov stability in discrete-time implementations:

**1. Projection methods and discrete gradient methods (Gonzalez, 1996; McLachlan & Quispel, 2002; Hairer et al., 2006).** These methods explicitly enforce the Lyapunov decrease condition $V(\mathbf{z}_{k+1}) < V(\mathbf{z}_k)$ at each step. Projection methods (Gonzalez, 1996) project the numerical solution onto sublevel sets of $V$, while discrete gradient methods (Itoh & Abe, 2001; McLachlan & Quispel, 2002) construct discrete analogues of the continuous gradient that preserve energy dissipation. However, both approaches require explicit knowledge of the Lyapunov function $V$ and its gradient, which may be computationally expensive to evaluate.

**2. Implicit Runge–Kutta methods (Hairer & Wanner, 1996; Ascher & Petzold, 1998).** These methods, particularly the Gauss and Radau families, guarantee stability for sufficiently small step sizes and only require access to the vector field $f$, without explicit knowledge of $V$. Under appropriate step size conditions, they preserve the asymptotic stability of equilibrium points (Stuart & Humphries, 1994; Hairer et al., 2006).

For our experiments, we use adaptive explicit Runge–Kutta methods (specifically `dopri5`) with appropriately chosen tolerances. While these do not provide strict Lyapunov preservation guarantees, extensive empirical validation shows that with sufficiently small error tolerances, the discrete trajectories closely approximate the continuous dynamics and maintain the convergence properties predicted by our theoretical analysis.

## F  Dataset Descriptions

**Two-dimensional spirals.**   The dataset consists of two coiling spirals corresponding to two distinct classes, each perturbed by Gaussian noise. Each spiral is a curve whose radius grows linearly with its polar angle, described in polar coordinates by $(\alpha\theta + \beta, \theta)$ with $\alpha = 2$ and $\beta = \pi$, which translates to Cartesian coordinates as

$$c(\theta) = ((2\theta + \pi)\cos(\theta), (2\theta + \pi)\sin(\theta)).$$

We define the two classes as:

$$S_1 = \{x \sim \mathcal{N}(c(\theta),\, I_d) \mid \theta \sim \text{Unif}[0, 2\pi]\}, \quad S_2 = \{x \sim \mathcal{N}(-c(\theta),\, I_d) \mid \theta \sim \text{Unif}[0, 2\pi]\}.$$

A total of 1000 samples (balanced between classes) are generated and split 70/30 into training and test sets.

The density admits a closed-form expression:

$$p(\mathbf{x}) = \frac{1}{2\sqrt{2\pi}} \int_0^{2\pi} e^{-\frac{\|\mathbf{x} - c(\theta)\|^2}{2}} + e^{-\frac{\|\mathbf{x} + c(\theta)\|^2}{2}}\, d\theta,$$

and when convolved with a Gaussian kernel of standard deviation $\sigma$, it becomes:

$$p_\sigma(\mathbf{x}) = \int_{\mathbf{x}' \in \mathbb{R}^2} \frac{1}{4\pi\sigma} e^{-\frac{1}{2}\frac{\|\mathbf{x} - \mathbf{x}'\|^2}{\sigma^2}}\, d\mathbf{x}' \int_0^{2\pi} e^{-\frac{\|\mathbf{x}' - c(\theta)\|^2}{2}} + e^{-\frac{\|\mathbf{x}' + c(\theta)\|^2}{2}}\, d\theta,$$

which is approximated numerically using the trapezoid method and differentiated via `autograd`.

**DermaMNIST.**   The DermaMNIST dataset (Yang et al., 2023) is a standardized, publicly available dataset designed for benchmarking machine learning models in the domain of dermatology. It is part of the broader MedMNIST collection, which encompasses various medical imaging datasets formatted similarly to MNIST. DermaMNIST focuses on skin lesion images derived from the HAM10000 dataset, consisting of 10,015 color dermatoscopic images of size $28 \times 28$ pixels (RGB). A total of 8,011 images form the training set, with the remainder forming the test set. The dataset includes seven classes representing different types of skin lesions:

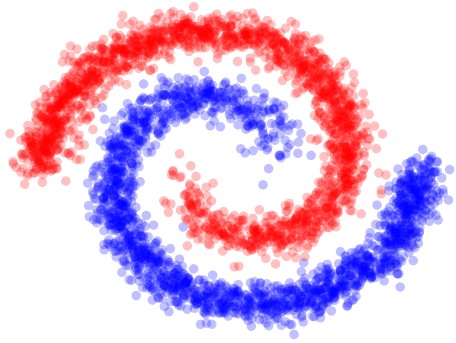

Figure 4: Two coiling spirals dataset.

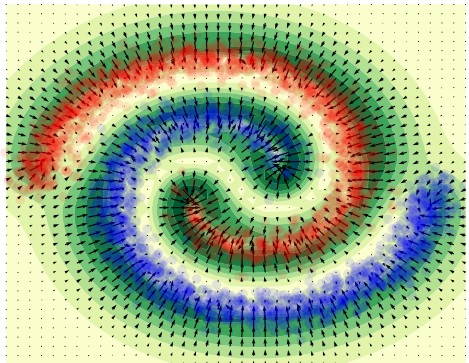

Figure 5: The two coiling spirals score field.

- Actinic keratoses and intraepithelial carcinoma / Bowen's disease (akiec)

- Basal cell carcinoma (bcc)

- Benign keratosis-like lesions (bkl)

- Dermatofibroma (df)

- Melanoma (mel)

- Melanocytic nevi (nv)

- Vascular lesions (vasc)

**PneumoniaMNIST.**   The PneumoniaMNIST dataset (Yang et al., 2023) focuses on binary classification of pneumonia from chest X-rays. It consists of grayscale images of size $28 \times 28$, derived from the Kaggle "Chest X-Ray Images (Pneumonia)" dataset. There are 5,232 training samples, with the remainder forming the test set. The dataset includes two classes:

- Normal: chest X-rays without signs of pneumonia

- Pneumonia: chest X-rays showing signs of pneumonia (bacterial or viral)

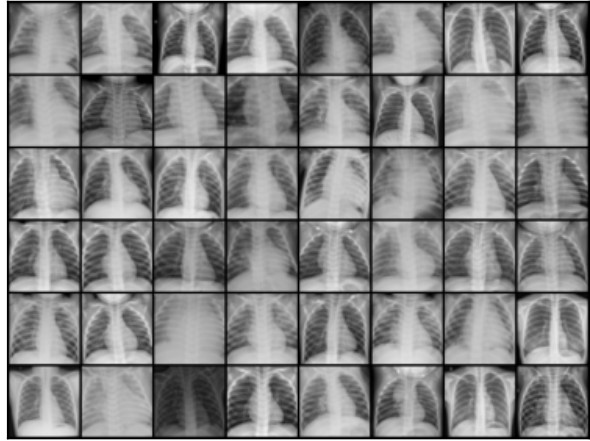

PneumoniaMNIST sample

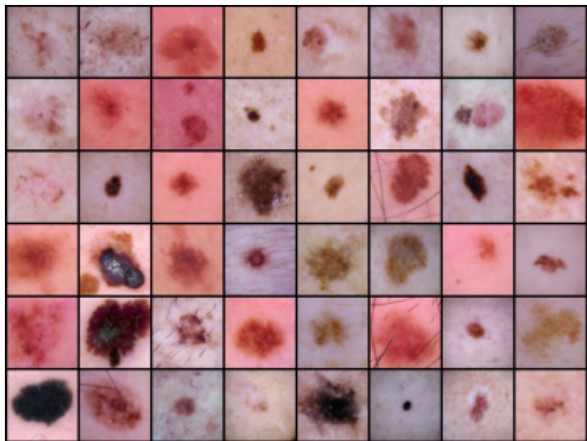

DermaMNIST sample

Figure 6: Sample images from the PneumoniaMNIST and DermaMNIST datasets, both downsampled to $28 \times 28$ pixels.

## G Adversarial Attack Configuration and Evaluation Metrics

We adopt an evaluation protocol similar to that of (Li et al., 2022c) and (Kang et al., 2021). All evaluated methods are tested against the FGSM and PGD attacks, with the PGD attack run for 20 iterations. The attack radius $\epsilon$ is set to 0.3 for the MNIST dataset and to $\frac{8}{255}$ for the CIFAR10, PneumoniaMNIST, and DermaMNIST datasets.

The choice of these specific attack radii follows standard practice in the robustness literature. In particular, $\epsilon = 0.3$ is widely used for MNIST, as it generates adversarial examples that meaningfully challenge model

robustness without producing unrealistic perturbations (Goodfellow et al., 2015). Similarly, $\epsilon = \frac{8}{255}$ is the canonical choice for CIFAR10 (**?**) and for other natural or medical image datasets of comparable resolution, such as PneumoniaMNIST and DermaMNIST (**?**). This ensures comparability with prior work and maintains a consistent perturbation scale across datasets.

We report two metrics:

- **Standard Accuracy:** classification accuracy on unperturbed (clean) test data.

- **Robust Accuracy:** classification accuracy under adversarial attacks, measuring the resilience of the model to small input perturbations.

## H   Additional Figures

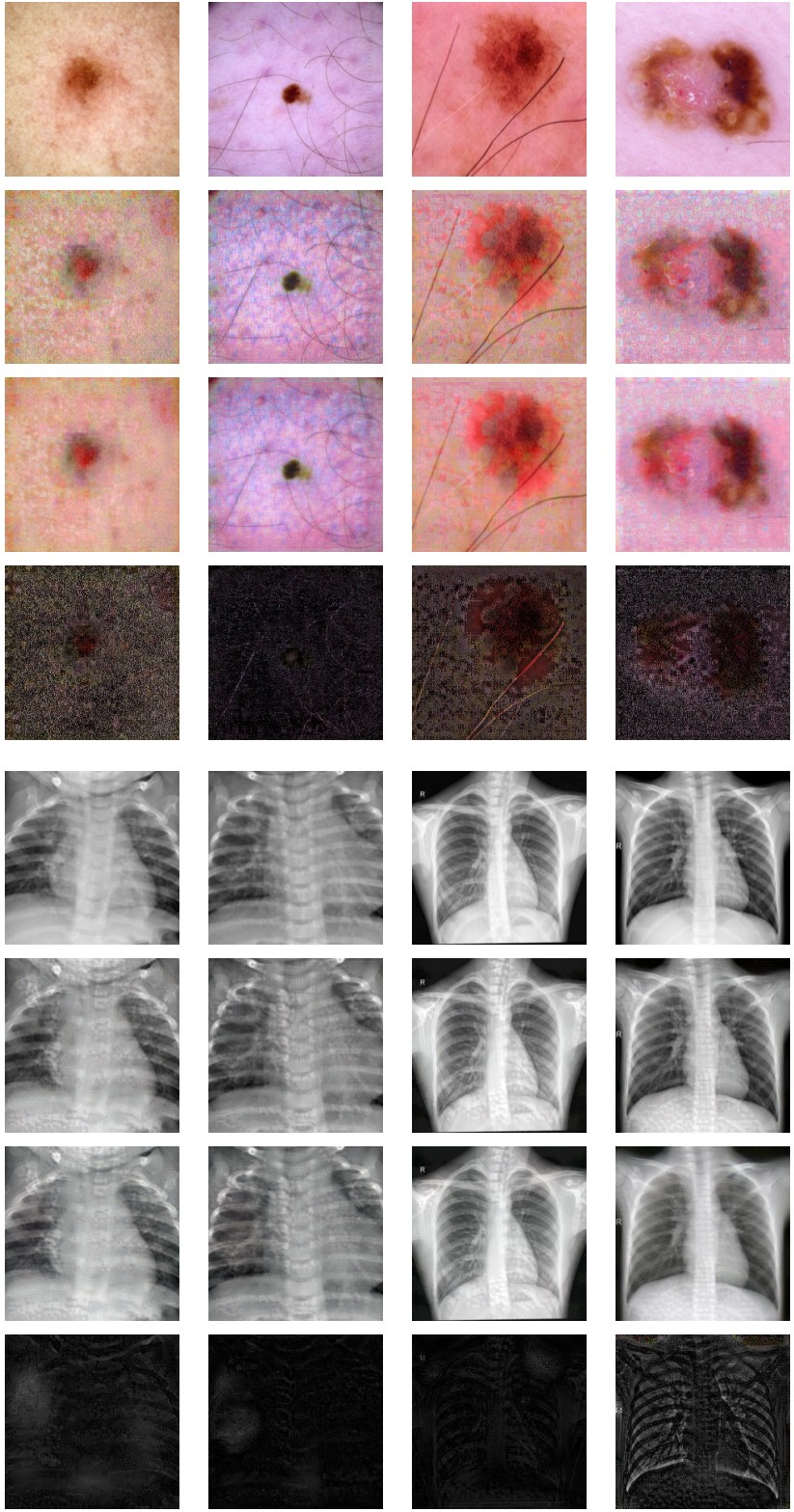

Figure 7: Illustration of the denoising effect achieved by our method on the DermaMNIST and PneumoniaMNIST datasets. The first row presents the original clean images, the second row displays the corresponding perturbed versions, the third row shows the outputs of our NeuralODE-based denoising layer applied to the perturbed images, and the fourth row depicts the pixel-wise difference between the denoised outputs and the original clean images. The last row, representing the residual, is computed in the HSV color space with the V (value) channel increased to enhance contrast for clearer visualization.

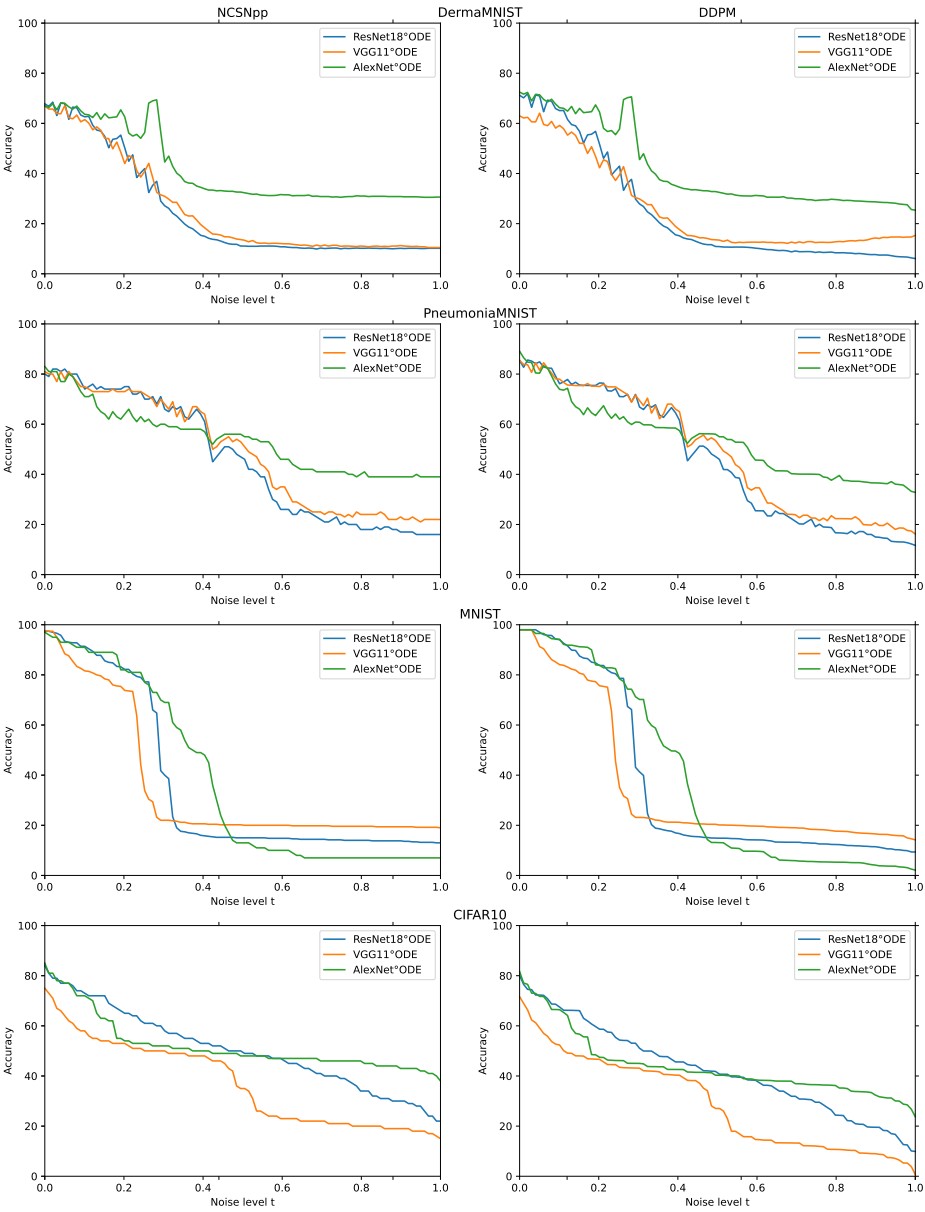

Figure 8: Robustness evaluation of the proposed method using three classifiers (ResNet18, VGG11, and AlexNet) across varying noise levels $\sigma$, tested on the DermaMNIST, PneumoniaMNIST, MNIST, and CIFAR-10 datasets.

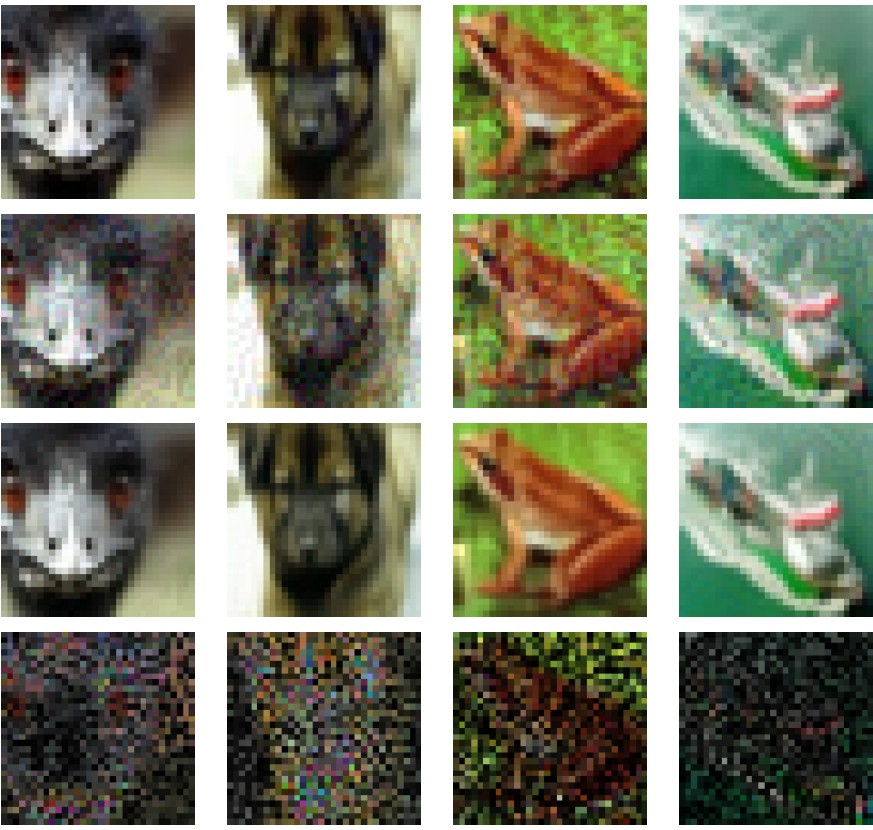

Figure 9: Illustration of the denoising effect achieved by our method on the CIFAR-10 dataset. The first row presents the original clean images, the second row displays the corresponding perturbed versions, the third row shows the outputs of our NeuralODE-based denoising layer applied to the perturbed images, and the fourth row depicts the pixel-wise difference between the denoised outputs and the original clean images. The last row, representing the residual, is computed in the HSV color space with increased brightness (V channel) to enhance visual contrast.

