# OpenReview forum: "Score-based Lyapunov Stable Neural ODE for Robust Classification"
_TMLR — Withdrawn by Authors_

### Review · Reviewer_5YKF · 2026-02-26

**Summary Of Contributions:**

The paper proposes an adversarial purification method, which (1) trains a score model via denoising score matching, (2) integrates the ODE to map the input to a purified point before classification.  The authors emphasise modularity and claim theoretical grounding via global convergence-to-critical-point results for gradient flows. Empirically, they report strong FGSM/PGD robustness on MNIST/CIFAR-10 and two MedMNIST datasets.

**Audience:**

Yes

**Audience Explanation:**

The findings of this paper are likely to be of interest to the safety and robustness community.

**Claims And Evidence:**

Yes

**Claims Explanation:**

1. The motivation and method are clear.

2. The authors provide sufficient evidence to support the effectiveness of their proposed method.

3. This paper is well-written, making it easy to follow.

**Requested Changes:**

1. It is unclear whether the reported FGSM and PGD adversarial examples are generated against the entire pipeline (ODE purification + classifier) or against the classifier only. If the attacks are conducted end-to-end, then achieving 70–90% accuracy under white-box attacks aises concerns about potential gradient masking. The paper should therefore include evaluations under stronger, adaptive attack protocols (AutoAttack, BPDA-PGD, and EOT-PGD) to substantiate robustness claims.

2. The paper should examine robustness under larger perturbation budgets, such as $\epsilon$ = 16/255 and 32/255.

3. Given the method’s plug-and-play nature, it would be informative to report performance when using a ViT-based classifier, in addition to CNN backbones.

4. The evaluation is limited to relatively small-scale datasets. Reporting results on a larger-scale benchmark (e.g., ImageNet) would better demonstrate practicality.

5. Since the method introduces an ODE integration step at inference time, it is important to report inference-time cost to clarify the robustness–efficiency trade-off.

---

### Review · Reviewer_hGAy · 2026-03-12

**Summary Of Contributions:**

The paper "Score-based Lyapunov Stable Neural ODE for Robust Classification" introduces a defense mechanism against adversarial attacks by treating neural networks as dynamical systems. The authors propose a two-stage architecture where an input first passes through a continuous-time Neural ODE layer before being fed into a standard classifier. This ODE layer acts as a powerful denoiser by implementing the gradient flow of the data distribution's score, naturally driving adversarially perturbed inputs toward asymptotically stable equilibrium points that correspond to the modes of the clean data.

**Audience:**

Yes

**Audience Explanation:**

The proposed modular architecture, which explicitly admits asymptotically stable equilibrium points, is a conceptually elegant approach to structural robustness. However, while the theoretical framework is promising, the manuscript currently suffers from significant limitations in experimental rigor, presentation clarity, and narrative consistency.

**Claims And Evidence:**

No

**Claims Explanation:**

The empirical validation is insufficient for modern adversarial robustness claims. The authors evaluate their defense using only FGSM and PGD-20. It is well-established in the robustness literature that PGD-20 is not a reliable standalone metric, as it often masks gradient obfuscation. The authors must evaluate their model using AutoAttack [1,2], which incorporates an ensemble of diverse parameter-free and black-box attacks, to provide a trustworthy assessment of both clean and robust accuracy.

[1] https://arxiv.org/abs/2003.01690
[2] https://arxiv.org/abs/2010.09670

**Requested Changes:**

(1) Figure 1 is quite abstract, and the main text surrounding it lacks necessary architectural details. While the classifier architectures are eventually detailed in Section 4.1, they should be referenced alongside the architectural overview for clarity. Furthermore, the authors highlight that the score model and classifier are trained independently. The manuscript would benefit from an explicit discussion confirming whether this implies the proposed ODE layer can be seamlessly integrated with any pre-trained, off-the-shelf model as a pure plug-and-play defense.

(2) A well-known drawback of continuous-time Neural ODEs and score-based methods is their computational overhead. The authors omit critical details regarding the training and inference costs. The manuscript must include a quantitative analysis of the computational budget (e.g., inference latency, FLOPs, or wall-clock time) compared to standard models and the chosen baselines.

(3) The authors only benchmark against other Lyapunov-based methods (e.g., SONet, SODEF, ASODE). Given that the proposed method cleverly sidesteps the need to compute higher-order information during training via denoising score matching, a direct comparison to standard Adversarial Training (AT) methods (e.g., standard PGD-AT or TRADES) is absolutely necessary to contextualize the method's effectiveness within the broader field.

(4) In the introduction, the authors criticize adversarial training for degrading clean accuracy—framing the robustness-accuracy tradeoff as a problem their structural approach aims to solve. Nevertheless, in the limitations section, they concede that their own method "introduces a slight reduction in clean accuracy due to the decoupling of the denoising and classification modules". The authors do not provide sufficient empirical evidence to prove that their method's tradeoff is actually superior to that of standard AT. This contradiction must be addressed, either by softening the initial claims or providing rigorous comparative evidence.

Minor Issues & Formatting

(a) Theoretical Flow: The mathematical presentation requires deep polishing. For example, Theorem 3.1, Proposition 3.2, and Corollary 3.3 are listed abruptly. The text lacks sufficient connective tissue explaining how these specific theoretical results translate practically into the proposed architecture.

(b) Implementation Details: The "Complete pipeline" is mentioned in only a brief four-line paragraph. This needs to be expanded into a formal algorithm block or a more detailed textual explanation to ensure reproducibility.

(c) Equation Numbering: Many equations throughout the manuscript (e.g., the score matching objectives) are not numbered. Please assign a number to every mathematical equation so that readers and reviewers can easily reference them.

---

### Review · Reviewer_PvPj · 2026-03-21

**Summary Of Contributions:**

This paper introduces a modular adversarial defense framework that prepends a Neural ODE-based purification layer to standard classifiers, mathematically connecting Lyapunov stability theory with score-based generative modeling to drive perturbed inputs toward high-density data modes. While the approach provides an elegant theoretical bridge and demonstrates improved empirical robustness against standard FGSM and PGD attacks compared to prior Lyapunov methods, its practical utility is currently limited by a slight degradation in clean accuracy, the absence of formal robustness guarantees, and the inherent computational overhead of using ODE solvers during inference.

**Additional Comments:**

N/A.

**Audience:**

Yes

**Audience Explanation:**

The findings will be of significant interest to the machine learning community, particularly researchers working at the intersection of adversarial robustness, Neural ODEs, and score-based generative models. The theoretical bridge established between Lyapunov stability and score estimation offers an elegant, principled framework for deterministic adversarial purification.

Even if the empirical evaluation requires more rigorous stress-testing against adaptive attacks, the conceptual contribution—along with its demonstrated potential in safety-critical domains like medical imaging—makes it a highly relevant and thought-provoking paper for theorists and security practitioners alike.

**Broader Impact Concerns:**

N/A.

**Claims And Evidence:**

No

**Claims Explanation:**

While most of the claims are supported by experiments, a critical evaluation is still missing. Architectures that rely on ODE solvers and multi-step sampling are notoriously prone to shattering or obfuscating backpropagated gradients. This causes standard PGD attacks to fail during optimization, resulting in a false sense of security (i.e., "obfuscated gradients" or "gradient masking"). The paper limits its evaluation to basic FGSM and standard PGD, completely omitting advanced evaluations like AutoAttack (the current gold standard for adversarial robustness) or adaptive attacks like BPDA (Backward Pass Differentiable Approximation). Without subjecting the model to these rigorous tests, the claims of substantial robustness improvements are entirely unsubstantiated.

**Requested Changes:**

1.A critical flaw in this approach is its vulnerability to basin crossing and score estimation errors, compounded by heavy inference delays. Because adversarial examples often lie in low-density regions where the score network ($\psi_\theta$) is inaccurate, the ODE trajectories can become highly unreliable. Worse, if a strong perturbation pushes an input slightly into a different class's basin of attraction, the gradient flow will actively "denoise" the sample into the wrong mode—effectively assisting the attacker. Finally, increasing the integration time ($t_f$) to achieve stronger convergence only worsens the model's computational cost.

2.The paper contains a direct and major contradiction regarding its implementation. In Section 4.1, the authors explicitly state they backpropagate through the stored computational graph “rather than using the adjoint method.” However, in Section 5, they claim their approach is memory-efficient precisely because it “leverages the adjoint method.” This blatant inconsistency undermines the paper's rigor and raises serious doubts about the actual memory footprint of the proposed architecture in practice.

---

### Author Response · Authors · 2026-04-04

We sincerely thank all reviewers for their detailed and constructive feedback.
We especially appreciate the emphasis on evaluating the proposed method under stronger adaptive attacks such as AutoAttack, BPDA, and EOT, as well as the insightful comments regarding computational cost, architectural clarity, and theoretical presentation.

Due to unexpected limitations in access to adequate computational resources during the rebuttal period, we were unfortunately unable to conduct the additional large-scale and adaptive experiments required to properly address these important concerns.

Given the significance of these points for validating the robustness claims, we believe that providing partial or incomplete answers would not meet the standards expected for this work.

Therefore, we intend to withdraw the current submission and revise the paper thoroughly, incorporating the reviewers’ suggestions, particularly a comprehensive evaluation under adaptive attacks and a clearer analysis of computational trade-offs, before resubmitting.

We are grateful for the reviewers’ valuable feedback, which will significantly help improve the quality and rigor of this work.

---

### Note · Authors · 2026-04-04

I have read and agree with the venue's withdrawal policy on behalf of myself and my co-authors.